# Ethnic inequalities in mental and physical multimorbidity in women of reproductive age: a data linkage cohort study

Raquel Catalao ,[1] Sarah Dorrington ,[1] Megan Pritchard,[2] Amelia Jewell ,[2] Matthew Broadbent,[2] Mark Ashworth ,[3] Stephani Hatch ,[1,4] Louise Howard [5]

¹Department of Psychological Medicine, King's College London, London, UK
²NIHR Biomedical Research Centre, South London and Maudsley NHS Foundation Trust, London, UK
³Primary Care and Public Health Sciences, King's College London, London, UK
⁴ESRC Centre for Society and Mental Health, King's College London, London, UK
⁵Section of Women's Mental Health, Institute of Psychiatry Psychology and Neuroscience, King's College London, London, UK

**Correspondence to**
Dr Raquel Catalao;
raquel.catalao@kcl.ac.uk

## ABSTRACT

**Objectives** Explore inequalities in risk factors, mental and physical health morbidity in non-pregnant women of reproductive age in contact with mental health services and how these vary per ethnicity.

**Design** Retrospective cohort study.

**Setting** Data from Lambeth DataNet, anonymised primary care records of this ethnically diverse London borough, linked to anonymised electronic mental health records ('CRIS secondary care database').

**Participants** Women aged 15–40 years with an episode of secondary mental health care between January 2008 and December 2018 and no antenatal or postnatal Read codes (n=3817) and a 4:1 age-matched comparison cohort (n=14 532).

**Main outcome measures** Preconception risk factors including low/high body mass index, smoking, alcohol, substance misuse, micronutrient deficiencies and physical diagnoses.

**Results** Women in contact with mental health services (whether with or without severe mental illness (SMI)) had a higher prevalence of all risk factors and physical health diagnoses studied. Women from minority ethnic groups were less likely to be diagnosed with depression in primary care compared with white British women (adjusted OR 0.66 (0.55–0.79) p<0.001), and black women were more likely to have a SMI (adjusted OR 2.79 (2.13–3.64) p<0.001). Black and Asian women were less likely to smoke or misuse substances and more likely to be vitamin D deficient. Black women were significantly more likely to be overweight (adjusted OR 3.47 (3.00–4.01) p<0.001), be diagnosed with hypertension (adjusted OR 3.95 (2.67–5.85) p<0.00) and have two or more physical health conditions (adj OR 1.94 (1.41–2.68) p<0.001) than white British women.

**Conclusions** Our results challenge the perspective that regular monitoring of physical health in primary care should be exclusively encouraged in people with a l diagnosis. The striking differences in multimorbidity for women in contact with mental health services and those of ethnic minority groups emphasise a need of integrative models of care.

## STRENGTHS AND LIMITATIONS OF THIS STUDY

⇒ Large sample of 18 349 women living in a ethnically diverse inner city borough.
⇒ Near complete coverage of primary care data in a London borough with good ethnicity recording allowed us to explore prevalence of multimorbidity in non-pregnant women of reproductive age at the population level, an often neglected population group.
⇒ Linkage with mental health care services records enabled us to explore disparities in women in contact with mental health services.
⇒ Clinicians may be biased in recording risk factors and physical health diagnoses as there are lack of incentives to record in those without severe mental illness.
⇒ We excluded highly mobile populations and defined ethnicity in five broader groups potentially obscuring differences within the groups.

## INTRODUCTION

People with mental disorders have high rates of physical comorbidities, which contribute to premature mortality and ongoing health inequalities.[1] Similarly, there is a well-established link between perinatal mental disorders and adverse pregnancy outcomes.[2] There are also striking inequalities in outcomes for black women during the perinatal period,[3] but few studies have focused on ethnic inequalities in physical and mental multimorbidities in women, nor on the prevalence of the conditions that affect women's reproductive health. However, improving health earlier in women's lives could have benefits for women, their children and the health of future generations.

In this study, we aimed to explore disparities in risk factors and physical health diagnoses in women of reproductive age in contact with

mental health (MH) services compared with primary care controls and how these may vary by ethnicity.

We hypothesised that women in contact with MH services, particularly those with severe mental illness (SMI; a diagnosis of schizophrenia, bipolar affective disorder and other psychoses) would have a higher prevalence of multimorbidity in primary care after controlling for deprivation, and women from ethnic minority groups would be disproportionally affected.

## METHODS
### Settings and data sources

Lambeth is an ethnically diverse borough, with a greater number of black Caribbean and black African residents, and fewer South Asian residents, than most other areas of London,[4] and a larger proportion of younger adults compared with averages in London and England.[5] Lambeth has high levels of deprivation[6] and high turnover.[5]

Anonymised primary care data were extracted on November 2020 from the computerised medical records of all general practices (n=48) as part of Lambeth DataNet (LDN). This data set includes data from all registered patients including diagnoses and prescriptions, stored as Read codes (a standard vocabulary for clinicians to record patient findings and procedures, in health and social care IT systems across UK primary and secondary care).[7]

Data were extracted from LDN for the study period January 2008–December 2018 (total number of registrations 855 742). Secondary care data came from the Clinical Record Interactive Search (CRIS),[8] an application allowing research access to pseudonymised electronic health record data from the South London and the Maudsley National Health Service (NHS) Foundation Trust (SLaM), Lambeth's MH provider.

### Data linkage

CRIS and LDN data were linked and stored by the SLaM Clinical Data Linkage Service, which provides a safe haven environment with strict governance arrangements. Data were linked using encrypted NHS numbers, which were subsequently removed and destroyed such that the linked dataset became fully anonymised.

### Inclusion criteria for exposed cohort

Women aged between 15 and 40 years at the start of the study window in contact with SLaM during the study window, defined as an active episode of care including a face-to-face event, were included in this study. Women were excluded from the exposed cohort if they had antenatal or postnatal Read codes in their primary care record before window end and were registered in LDN for less than 2 years during in the study window. At least 1 year registration in LDN occurred after the earliest ever SLaM start date irrespective of study window.

### Inclusion criteria for unexposed cohort

Exact age 4:1 matched women from LDN with at least 2 years' registration on LDN during study window, no antenatal nor postnatal related Read codes before window end and no SLaM MH care receipt.

### Measures
#### Age

Age at start of the study (ie, 1 January 2008) was calculated from subtracting the year of birth from year of data extraction.

#### Ethnicity

Ethnicity was extracted from LDN self-reported ethnicity data fields. There were over 100 different codes for ethnicity, and these were broadly categorised in five different groups (white British; white other; black; Asian and mixed/other), as defined in 2011 Census.[4]

#### Index of Multiple Deprivation (IMD)

Information on 768 Lower Super Output Areas (geographical areas of around 650 households) associated with earliest LDN registration in the window was used to determine level of social deprivation using the IMD-2019.[9] The IMD combines information from seven domains (income deprivation, employment, education skills and training, health deprivation and disability, crime, barriers to housing and services and living environment deprivation) to produce an overall relative measure of deprivation.

#### MH diagnoses

Data extracted from LDN included SMI status using Quality and Outcomes Framework criteria, which is an annual reward and incentive programme for all primary care practices in England.[10] Data extracted from CRIS included ICD-10 (International Classification of Diseases-10) diagnosis consistent with the same SMI diagnoses: F20, F22, F23, F25, F28 and F29 (schizophrenia and related disorders, schizoaffective disorders and delusional disorders); F30 and F31 (mania and bipolar affective disorders); and F32.3 and F33.3 (psychotic depression). Thus, a subgroup of the exposed cohort was defined as SMI if they were part of the SMI register on LDN or had an ICD-10 diagnosis of SMI recorded on CRIS.

Data extracted from LDN also included those on the Depression register, which is defined as patients who have an unresolved record of depression in their clinical record.[10]

### Physical health variables derived from Read codes

The following Read codes were extracted if present during study window:

#### Body mass index (BMI)

Women were defined as overweight during the study window if they had BMI value recorded of over 25.0 or had Read codes in register associated with BMI above 25 or obesity. Women were defined as underweight when they had a BMI <18.5 or Read codes in their register consistent with an anorexia nervosa diagnosis. When several BMI values were available, we used the most recent.

### Smoking
Read codes for current smoker.

### High-risk alcohol use
Alcohol use disorders identification test consumption (AUDIT-C) questionnaire score ≥5 or Read codes for hazardous, harmful alcohol use or alcohol dependence

### Drug use
Read codes for substance misuse.

### Folic acid prescription
Extracted via Dictionary of Medicines and Devices (DMD) codes[11] for all folic acid formulary preparations.

### Vitamin D deficiency
Read codes for vitamin D deficiency.

### Emergency and long-acting reversible contraception (LARC)
DMD and relevant Read codes for LARC and emergency contraception prescriptions.

### Physical health diagnoses
Read codes for diagnoses including asthma, hypertension, diabetes, epilepsy, HIV, hepatitis B and C, endometriosis and polycystic ovarian syndrome (PCOS).

### Psychiatric drugs
DMD codes on LDN and prescriptions from CRIS for Valproate, Lithium, antidepressant and antipsychotic medication.

### Healthcare use
Number of LDN recorded primary care consultations included consultations with a doctor or nurse as face-to-face GP appointments, emergency consultations, home visits, out of hours or walk in clinics. Number of CRIS recorded psychiatric inpatient days and number of face-to-face attended community contacts.

### Mortality
Number of deaths was ascertained by calculating number of people with a date of death extracted from LDN at the time of data extraction.

### Data analysis
Data were analysed in Stata V.15.0 for Mac after cleaning. Descriptive statistics were used to look for differences in demographic factors between exposed cohort and unexposed cohort.

Univariate and multivariate logistic and negative binomial regression analyses (adjusting for a priori determined confounders of IMD and ethnicity) were used to examine the rate of risk factors, diagnoses and health services contacts across groups (controls vs women in contact with MH services) and a subgroup analysis within the exposed group (SMI vs non SMI diagnoses). A complete case analysis was also conducted to explore disparities in the outcomes above between different ethnic groups in primary care with adjustment for IMD, SMI diagnoses and age.

### Patient and public involvement
The authors had no direct contact information of the study participants because anonymised clinical data were used in accordance with strict confidentiality guidelines. All patients have the choice to opt-out of their anonymised data being used. CRIS and LDN were developed with extensive service user involvement and adheres to strict governance frameworks managed by service users. No patient was involved in developing the hypothesis, research questions, plans for the study's design or writing of the results. Results will be disseminated to local public health structures including Lambeth Health Watch.

## RESULTS
There were 3817 women from LDN who were in contact with MH services (figure 1). These women were more likely to be from black ethnic groups (OR 1.48 (1.31–1.67) p<0.01) and live in areas of higher deprivation (OR 1.01 (1.01–1.02) p<0.001) compared with the rest of the LDN cohort. Six hundred and eighty-eight (18.0%) of the women in contact with mental care had a recorded SMI diagnosis; 420 (71.7%) of these women had recorded diagnoses of SMI on both LDN and CRIS; 102 (14.8%) women had a SMI diagnosis recorded on CRIS only; and 166 women (24.1%) had SMI diagnosis recorded on LDN only. Subgroup analysis showed that women with SMI were more likely to be older and live in areas of higher deprivation (table 1). Women with SMI diagnosis recorded on LDN but not on CRIS had higher odds of substance misuse but not multimorbidity.

### Missing data
Around 17.4% (n=2535) of women in the control group and 323 (8.5%) women in contact with MH services had missing data on ethnicity. Six hundred and twenty-one (4.3%) of women in the control group and 104 women (2.7%) in women in contact with MH services had missing data on IMD (table 1).

Missing data on ethnicity were associated with not being in contact with MH services, older age, living in areas of less deprivation and being a non-smoker; missing data on IMD were associated with younger age and Asian ethnicity.

Five hundred and thirty-two (13.9%) women in contact with MH services and 4582 (31.5%) in the comparison group had missing data on BMI; this was associated with not being in contact with MH services, older age and being of an ethnicity other than white British.

### Inequalities in health outcomes
#### Women in contact with MH services versus primary care contact only
Women in contact with MH services had a higher prevalence of all risk factors investigated (tables 2 and 3). They

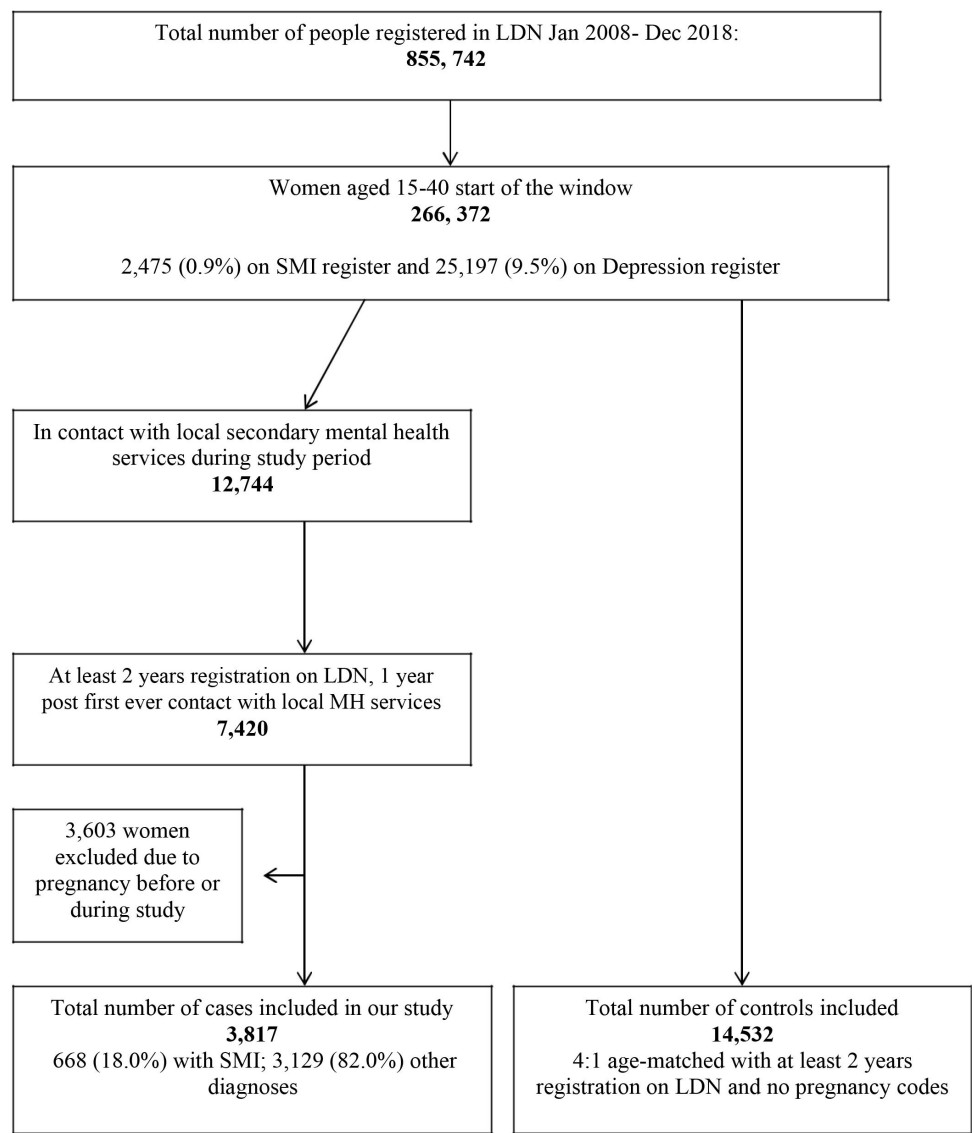

**Figure 1** Exposed cohort and controls selection from Lambeth DataNet (LDN).

were more likely to have a weight in the abnormal range: BMI >25 (adj OR 1.46 (1.33–1.60) p<0.0001) or BMI <18.5 (adj OR 2.39 (2.05–2.19) p<0.001) (table 3). They had a higher prevalence of alcohol (adj OR 10.39 (8.10–13.33) p<0.001) and substance misuse (adj OR 39.37 (23.87–64.86) p<0.001) after adjustment for ethnicity and IMD. They had higher rates of vitamin D deficiency, emergency contraception use and termination of pregnancy.

Women in contact with MH services had a higher prevalence of physical multimorbidity (two or more physical health disorders) (9.0% vs 1.4%; adjusted OR 2.93 (2.37–3.63) p<0.001). They had a particularly high prevalence of epilepsy (adjusted OR 5.73 (4.00–8.20) p<0.001), HIV (adj OR 4.77 (2.66–8.56) p<0.001) and chronic hepatitis (adj OR 8.24 (5.12–13.24) p<0.001) (tables 2 and 3).

Only 28.4% of all patients prescribed valproate had a folic acid prescription issued in primary care. Similarly whereas women on valproate were more likely to be given LARC in primary care (OR 2.27 (1.40–3.68) p<0.001), only 10.6% of women on valproate were on LARC.

Women in contact with MH services had a higher number of face-to-face clinical consultations in primary care during study period (adjusted IRR 2.74 (2.62–2.87) p<0.001) after adjusting for sociodemographic factors, SMI and a diabetes diagnosis. They were approximately 16 times more likely to die (adjusted OR 16.10 (8.37–30.91) p<0.001) after adjusting for IMD and ethnicity, and most deaths were among women without a SMI diagnosis (tables 1 and 3).

### Subgroup analysis: SMI versus non-SMI diagnoses in woman in contact with MH services

Women with SMI were more likely to be overweight (adjusted OR 1.80 (1.48–2.19) p<0.001); smokers (adjusted OR 1.41 (1.18–1.69) p<0.001); have a diagnosis of diabetes (adj OR 2.31 (1.58–3.36) p<0.001) and hypertension (adj OR 1.71 (1.15–2.56) p<0.001) than other women in contact with MH services with no SMI diagnoses after adjusting for IMD and ethnicity (online supplemental file 1). They were also more likely to have

**Table 1** Characteristics of women of reproductive age in our sample: by contact with mental health services

| | Primary care controls (n=14 532) | Missing | In contact with local MH services (n=3817) | | Missing |
| | | | No SMI (n=3129) | SMI diagnoses (n=688) | |
|---|---|---|---|---|---|
| Age | | 0 (0.0%) | | | 0 (0.0%) |
| 15–19 | 3030 (20.9%) | | 762 (25.7%) | 111 (16.1%) | |
| 20–24 | 3430 (23.6%) | | 718 (24.2%) | 128 (18.6%) | |
| 25–29 | 2706 (18.6%) | | 541 (18.2%) | 115 (16.7%) | |
| 30–34 | 2312 (15.9%) | | 442 (14.9%) | 130 (18.9%) | |
| 35–40 | 3054 (21.0%) | | 506 (17.0%) | 204 (29.6%) | |
| Ethnicity | | 2535 (17.4%) | | | 323 (8.5%) |
| White British | 2467 (20.6%) | | 698 (24.5%) | 92 (14.3%) | |
| Other white | 4615 (38.5%) | | 736 (25.8%) | 152 (23.6%) | |
| Black | 1392 (11.6%) | | 452 (15.9%) | 206 (32.0%)) | |
| Asian | 830 (6.9%) | | 149 (5.2%) | 32 (5.0%) | |
| Other | 2693 (22.5%) | | 816 (28.6%) | 161 (25.0%) | |
| IMD score | | 621 (4.3%) | | | 104 (2.7%) |
| First quintile | 2211 (15.9%) | | 561 (18.4%) | 153 (23.0%) | |
| Second quintile | 6615 (47.6%) | | 1490 (48.9%) | 325 (48.9%) | |
| Third quintile | 3783 (27.2%) | | 791 (26.0%) | 158 (23.8%) | |
| Fourth quintile | 1078 (7.8%) | | 168 (5.5%) | 27 (4.1%) | |
| Fifth (least deprived) | 224 (1.6%) | | 38 (1.3%) | 2 (0.3%) | |
| Inpatient during study period ($\chi^2$) | – | | 184 (5.9%) | 305 (44.3%)*** | |
| Number of inpatient days | | | | | |
| Median test | – | | Median 0 IQR (0–0) | Median 0 IQR (0–61)*** | |
| Number of days with attended CMHT f2f contacts | | | | | |
| Median test | – | | Median 5 IQR (1–21) | Median 37 IQR (7–111)*** | |
| Number of primary care clinical contacts | | | | | |
| Median test | Median 13 IQR (2–34) | | Median 59 IQR(27-110) | Median 89 IQR(37 – 157)*** | |
| Mortality | 18 (0.1%) | | 45 (1.4%) | 15 (2.2%) | |

***P<0.001, **p<0.01 and *p<0.05.
MH, mental health; SMI, severe mental illness.

a higher number of attended community MH face-to-face appointments, inpatient days in MH services, as well as GP face-to-face consultations during the study period.

### Racial and ethnic disparities
#### Mental health
Across cases and controls with data on ethnicity (n=15 491), there were significant discrepancies in diagnosis of depression in primary care. Fifteen per cent of the whole sample (n=2688) was on the primary care depression register. Women from white other (adj OR 0.49 (0.44–0.56) p<0.001), Asian (adj OR 0.53 (0.43–0.65) p<0.001) and black groups (adj OR 0.67 (0.57–0.78) p<0.001) were less likely to be diagnosed with depression in primary care compared with white British women after adjustment for age and IMD. Conversely, black women were more likely to have a SMI diagnosis (adjusted OR 2.79 (2.13–3.64) p<0.001) (tables 4 and 5).

#### Risk factors
Black women (adjusted OR 3.47 (3.00–4.01) p<0.001) and women from mixed/other ethnicities (adjusted OR 1.24 (1.10–1.40) p<0.001) were significantly more likely to be overweight after adjusting for SMI status, IMD and age and black, Asian and women from mixed/other ethnicities were also more likely to have a BMI below 18.5 compared with white British women (table 5 and online supplemental file 2).

Both black and Asian women were less likely to smoke, use alcohol excessively or misuse substances than white British women.

Black (adjusted OR 3.03 (2.38–3.87) p<0.001), Asian (adjusted OR 2.72 (2.04–3.64) p<0.001) and women from mixed and other ethnicities (adj OR 1.40 (1.09–1.78) p<0.001) were more likely to be vitamin D deficient. They

**Table 2** Prevalence of risk factors and physical health disorders by contact with MH services and SMI status

| Total=18 349 | Controls (n=14 532), n (%) | In contact with MH services (n=3817), n (%) | |
|---|---|---|---|
| | | No SMI (n=3129) | SMI diagnoses (n=688) |
| Overweight (BMI >25) n=13 235 | 3086 (31.0) | 963 (36.3) | 343 (54.4) |
| Underweight (BMI <18.5) n=13 235 | 556 (5.6) | 299 (11.3) | 48 (7.6) |
| Smoker | 3010 (20.7) | 1322 (42.3) | 333 (48.4) |
| Excessive alcohol use | 92 (0.6) | 230 (7.4) | 45 (6.4) |
| Drug use | 19 (0.1) | 166 (5.3) | 38 (5.5) |
| Vitamin D deficiency | 469 (3.2) | 324 (10.4) | 84 (12.2) |
| Vitamin D prescription | 632 (4.4) | 525 (16.8) | 126 (18.3) |
| Folate prescription | 294 (2.0) | 230 (7.4) | 122 (17.7) |
| LARC use | 655 (4.5) | 280 (8.9) | 50 (7.3) |
| Emergency contraception | 374 (2.6) | 239 (7.6) | 55 (8.0) |
| TOP | 153 (1.0) | 89 (2.8) | 11 (1.6) |
| Psychiatric medication prescribed | | | |
| Valproate | 12 (0.1) | 29 (0.9) | 129 (18.8) |
| Antidepressants | 1467 (10.1) | 2045 (65.4) | 470 (68.3) |
| Antipsychotics | 51 (0.4) | 302 (9.7) | 536 (77.9) |
| Physical health diagnoses | | | |
| Asthma | 851 (5.9) | 427 (13.7) | 91 (13.%) |
| Diabetes | 174 (1.2) | 93 (3.0) | 53 (7.7) |
| Hypertension | 258 (1.8) | 91 (3.0) | 45 (6.5) |
| Cardiovascular disease | 27 (0.2) | 7 (0.2) | 5 (0.7) |
| Epilepsy | 50 (0.3) | 77 (2.5) | 18 (2.6) |
| Endometriosis | 156 (1.1) | 73 (2.3) | 9 (1.3) |
| PCOS | 370 (2.6) | 174 (5.6) | 27 (3.9) |
| HIV | 19 (0.1) | 29 (0.9) | 4 (0.6) |
| Hepatitis B/C | 26 (0.2) | 48 (1.5) | 12 (1.7) |
| Multimorbidity (two or more long-term conditions) | | | |
| Physical | 206 (1.4) | 269 (8.6) | 74 (10.8) |
| Mental and physical | 395 (2.7) | 660 (21.1) | 428 (62.2) |

BMI, body mass index; LARC, long-acting reversible contraception; MH, mental health; PCOS, polycystic ovarian syndrome; SMI, severe mental illness; TOP, termination of pregnancy.

were also more likely to be prescribed folic acid after adjustment for SMI, valproate use and age (table 5).

LARC prescription was low in our sample with less than 10% of women obtaining this method in primary care. Women from all ethnic minority groups were less likely to be prescribed or administered LARC in primary care compared with white British women, uptake was particularly low in Asian women (adjusted OR 0.36 (0.24–0.54) p<0.001). Women from black ethnic groups were also more likely to visit GP for emergency contraception (adjusted OR 1.64 (1.27–2.12) p<0.001) and termination of pregnancy (adjusted OR 1.68 (1.10–2.56) p=0.016).

### Psychiatric medication

Black, Asian and women from white other ethnic groups were less likely to be prescribed antidepressant medication in primary care compared with white British women, whereas black women were more likely to be prescribed antipsychotic medication after adjustment for SMI diagnoses, IMD and age.

### Physical health diagnoses

Asthma was the most prevalent (7.5% of total sample) physical health diagnosis in all ethnic groups except in black women where hypertension was the most prevalent diagnosis (9.4% of all black women) (table 4).

Black women were almost four times more likely to be diagnosed with hypertension than white British women (adjusted OR 3.95 (2.67–5.85) p<0.001). Diabetes was also more prevalent among black (adjusted OR 2.26 (1.51–3.40) p<0.001) and Asian women (adjusted OR 2.20 (1.34–3.60) p<0.001) compared with white British women (table 5).

The prevalence of PCOS and endometriosis was 3.1% and 1.3% of total sample, respectively, and there were no differences among different ethnic groups.

The numbers of women with HIV and hepatitis B/C in our sample were too small to conduct multivariate logistical regression analyses, but the prevalence of HIV was disproportionally higher in black women (1.7%) compared with whole sample prevalence (0.3%) (table 4).

Overall, black women had higher rates of physical multimorbidity compared with white British women (prevalence 6.3% vs 2.1%; adjusted OR 1.94 (1.41–2.68) p<0.001) (tables 4 and 5).

### Healthcare contacts

Black women had a higher number of face-to-face primary care appointments during the study period than women from other ethnic groups after adjustment for IMD, SMI diagnoses and age (adjusted IRR 1.26 (1.18–1.35) p<0.001) (table 5).

After adjustment for SMI and IMD results showed that all ethnic minority groups had less community MH face-to-face contacts compared with white British women.

Of the 15 491 women with data on ethnicity, there were 69 (0.5%) deaths. The numbers were too small to perform logistical regression per ethnic group, but there were significant differences with black women having the higher risk of death (n=20, 1.0% Fisher's exact test p<0.001) (table 4).

## DISCUSSION
### Principal findings

In this study using a primary care - secondary MH care clinical data linkage of an ethnically diverse and deprived inner London borough, we found striking differences in risk factors, multimorbidity and health service contacts by ethnicity and MH service use for women of reproductive age. Women in contact with MH services had

**Table 3** Association between contact with local MH services and risk factors, diagnoses and clinical contacts

| | In contact with MH services Univariate logistical regression OR, 95% CI (n=18 349) | Multivariate logistic regression Adjusted model for ethnicity and IMD Adjusted OR, 95% CI (n=14 980) |
|---|---|---|
| Overweight | 1.64 (1.50 to 1.78)*** n=13 235 | 1.46 (1.33 to 1.60)*** n=12 116 |
| Underweight | 2.41 (2.09 to 2.79)*** n=13 235 | 2.46 (2.09 to 2.89)*** n=12 116 |
| Smoking | 2.93 (2.72 to 3.16)*** | 2.78 (2.56 to 3.02)*** |
| Alcohol abuse | 12.19 (9.60 to 15.47)*** | 10.39 (8.10 to 13.33)*** |
| Drug use | 43.13 (26.91 to 69.11)*** | 39.35 (23.87 to 64.86)*** |
| Folate prescription | 4.92 (4.19 to 5.77)*** | 4.05 (3.42 to 4.80)*** |
| Vitamin D deficiency | 3.58 (3.12 to 4.11)*** | 2.92 (2.52 to 3.39)*** |
| Vitamin D prescription | 4.52 (4.03 to 5.08)*** | 3.71 (3.27 to 4.21)*** |
| LARC | 2.01 (1.75 to 2.30)*** | 1.67 (1.44 to 1.94)*** |
| Emergency contraception | 3.16 (2.70 to 3.70)*** | 2.70 (2.28 to 3.19)*** |
| TOP | 2.52 (1.96 to 3.26)*** | 2.10 (1.60 to 2.76)*** |
| **Physical health diagnoses** | | **Adjusted model for ethnicity and IMD Adj OR, 95% CI (n=11 989)** |
| Asthma | 2.52 (2.25 to 2.83)*** | 2.08 (1.84 to 2.35)*** |
| Diabetes | 3.28 (2.62 to 4.10)*** | 2.55 (2.01 to 3.24)*** |
| Hypertension | 2.07 (1.68 to 2.56)*** | 1.39 (1.11 to 1.74)* |
| Epilepsy | 7.39 (5.24 to 10.43)*** | 5.73 (4.00 to 8.20)*** |
| PCOS | 2.13 (1.78 to 2.53)*** | 1.75 (1.31 to 2.32)*** |
| Endometriosis | 2.02 (1.54 to 2.65)*** | 1.87 (1.55 to 2.25)*** |
| HIV | 6.66 (3.78 to 11.73)*** | 4.77 (2.66 to 8.56)** |
| Hepatitis B/C | 8.91 (5.62 to 14.14)*** | 8.24 (5.12 to 13.24)*** |
| Mortality | 12.88 (7.60 to 21.83)*** | 16.10 (8.37 to 30.91)*** |
| Multimorbidity | | |
| Physical | 4.09 (3.34 to 5.00)*** | 2.93 (2.37 to 3.63)*** |
| Mental and physical | 14.3 (12.6 to 16.1)*** | 11.7 (10.3 to 13.3)*** |
| **Health contacts** | **Univariate negative binominal regression IRR, 95% CI (n=18 165)** | **Adjusted model for SMI, age, IMD, ethnicity and diabetes Adjusted IRR, 95% CI (n=14 841)** |
| GP consultations | 3.47 (3.31 to 3.64)*** | 2.92 (2.80 to 3.05)*** |

***P<0.001, **p<0.01, *p<0.05.
IMD, Index of Multiple Deprivation; IRR, incidence rate ratio; LARC, long-acting reversible contraception; MH, mental health; PCOS, polycystic ovarian syndrome; SMI, severe mental illness.

higher prevalence of all physical health diagnoses and risk factors studied including a BMI outside the healthy range, smoking, alcohol and substance misuse as well as micronutrient deficiencies. They also had higher rates of emergency contraception use and termination of pregnancy stressing a greater unmet need for contraception. Similarly to previous studies, we found the risk of HIV and chronic hepatitis[12] to be greatly elevated compared with controls. Adjusting for hypothesised exploratory factors including ethnicity, deprivation and SMI diagnosis only partly explained inequalities. The risk of all-cause mortality in this group was 16 times higher compared with controls and, notably, women without an SMI diagnosis were also at increased risk of death. Black women and women in contact with MH services had a higher number of face-to-face clinical contacts in primary care so inequalities cannot be exclusively explained by lack of access.

### Comparison with other studies
Our study adds to growing literature of inequalities in mental and physical health outcomes for women of ethnic minority groups in the UK. Our results echo studies during the perinatal period that found women from ethnic minority groups are less likely to be diagnosed and receive treatment for depression in primary care,[13] and paradoxically black women are more likely to have an SMI diagnosis.[14] We found that black and Asian women were less likely to engage in some risk behaviours, such as smoking or drinking alcohol excessively compared

**Table 4** Prevalence of mental and physical health diagnoses, risk factors and clinical contacts by ethnicity in total sample

| Total (n=15 491) | White British (n=3357, 21.1%) | White other (n=5503, 35.5%) | Black (n=2050, 13.2 %) | Asian (n=1011, 6.5%) | Other (n=3670, 23.7%) |
|---|---|---|---|---|---|
| SMI, n (%) | 92 (2.8) | 152 (2.8) | 206 (10.1) | 32 (3.2) | 161 (4.4) |
| Depression, n (%) | 660 (20.3) | 659 (12.0) | 361 (17.6) | 126 (12.5) | 748 (20.4) |
| Asthma, n (%) | 320 (9.8) | 303 (5.5) | 193 (9.4) | 73 (7.2) | 411 (11.2) |
| Diabetes, n (%) | 35 (1.1) | 62 (1.1) | 111 (5.4) | 32 (3.2) | 57 (1.6) |
| Hypertension, n (%) | 34 (1.0) | 67 (1.2) | 192 (9.4) | 20 (2.0) | 61 (1.7) |
| Epilepsy, n (%) | 32 (1.0) | 23 (0.4) | 24 (1.8) | 8 (0.8) | 50 (1.4) |
| Endometriosis, n (%) | 40 (1.2) | 58 (1.1) | 44 (2.2) | 15 (1.%) | 69 (1.9) |
| PCOS, n (%) | 131 (4.0) | 154 (2.8) | 62 (3.0) | 32 (3.2) | 142 (3.9) |
| HIV, n (%) | 2 (0.1) | 8 (0.2) | 35 (1.7) | 0 (0.0) | 6 (0.2) |
| Hepatitis B/C, n (%) | 17 (0.5) | 27 (0.5) | 15 (0.7) | 6 (0.6) | 21 (0.6) |
| Overweight (BMI >25) n=12 353, n (%) | 730 (26.0) | 1183 (28.0) | 1012 (62.2) | 206 (27.1) | 989 (32.3) |
| Underweight BMI <18.5 n=12 353, n (%) | 172 (6.1) | 295 (7.0) | 88 (5.4) | 82 (10.8) | 209 (6.8) |
| Smoker, n (%) | 921 (28.3) | 1685 (30.6) | 414 (20.2) | 165 (16.3) | 1157 (31.5) |
| Alcohol abuse, n (%) | 107 (3.3) | 83 (1.5) | 33 (1.7) | 9 (1.1) | 114 (2.8) |
| Drug use, n (%) | 59 (1.8) | 58 (1.1) | 44 (2.2) | 15 (1.5) | 69 (1.9) |
| Vitamin D deficiency, n (%) | 109 (3.4) | 167 (3.0) | 269 (13.1) | 94 (9.3) | 189 (5.2) |
| LARC use, n (%) | 265 (8.1) | 264 (4.8) | 119 (5.8) | 31 (3.1) | 238 (6.5) |
| EHC, n (%) | 132 (4.1) | 150 (2.7) | 139 (6.8) | 26 (2.6) | 170 (4.6) |
| TOP, n (%) | 51 (1.6) | 51 (0.9) | 48 (2.3) | 23 (2.3) | 63 (1.7) |
| Antipsychotic use, n (%) | 135 (4.1) | 181 (3.3) | 237 (11.6) | 44 (4.4) | 227 (6.2) |
| Antidepressant use, n (%) | 903 (27.7) | 981 (17.8) | 596 (29.1) | 178 (17.6) | 1035 (28.2) |
| Physical multimorbidity, n (%) | 68 (2.1) | 68 (1.2) | 130 (6.3) | 26 (2.6) | 81 (2.2) |
| Physical and mental multimorbidity, n (%) | 286 (8.8) | 286 (5.2) | 342 (16.7) | 72 (7.1) | 419 (11.4) |
| Number of contacts in primary care | Median 26 IQR 12–54) | Median 14 IQR (4–38) | Median 43 IQR (12–101) | Median 18 IQR (5–53) | Median 28 IQR (11–60) |
| Number of contacts in CMHTS | Median 9 IQR (1– 30) | Median 4 IQR (1– 25) | Median 10 IQR (1– 43) | Median 5 IQR (1–24) | Median 7 IQR (1–32) |
| Mortality | 21 (0.6%) | 15 (0.3%) | 20 (1.0%) | 5 (0.5%) | 8 (0.2%) |

BMI, body mass index; CMHTS, community mental health teams; EHC, emergency contraception use; LARC, long-acting reversible contraception; PCOS, polycystic ovarian syndrome; SMI, severe mental illness.

with white British but were more likely to have unhealthy weight. Black women were also more likely to be prescribed antipsychotics irrespective of diagnosis. There were also inequalities in sexual and reproductive health and women from ethnic minority groups had decreased uptake of LARC in primary care. Consistent with previous national survey data,[15] black women had higher use of emergency contraception than white British women. We also confirmed that black women were at disproportionally high risk of hypertension compared with white British women.[16]

## Strengths and weaknesses

To our knowledge, this is one of the few studies investigating multimorbidity in non-pregnant women of reproductive age at the population level and highlighting disparities by ethnicity and mental ill-health, with a large sample size and near complete coverage of primary care

data in a London borough. Limitations include the exclusion of highly mobile populations not in contact with the same practice for 2 years, who may be at higher risk of poor health and under recording of risk factors and diagnoses (potentially due to clinicians' unconscious biases and lack of incentives to record risk factors in patients without SMI). Previous studies have found differences in multimorbidity within the broader ethnic groups defined in this study (eg, black Caribbean and black African),[17] but further disaggregation was not possible due to small cells. Our data on use of contraception must also be interpreted with caution, as these are patterns of use and access in primary care, whereas there are alternative sources of these services in community sexual health clinics and pharmacies. National data have shown that women living in urban areas and from ethnic minority groups are more likely to visit sexual health clinics compared

**Table 5** Association between ethnicity, risk factors, health diagnoses and clinical contacts

| Base (white British) | Multivariate logistical regressions Adjusted OR; 95% CI Model adjusted for IMD and age (ordinal) (n=14 980) | | | |
|---|---|---|---|---|
| | White other | Black | Asian | Other |
| SMI | 0.88 (0.64 to 1.09) | 2.79 (2.13 to 3.64)*** | 1.07 (0.71 to 1.61) | 1.42 (1.09 to 1.85)** |
| Depression | 0.49 (0.44 to 0.56)*** | 0.67 (0.57 to 0.78)*** | 0.53 (0.43 to 0.65)*** | 0.95 (0.84 to 1.07) |
| Model adjusted for IMD, SMI and age (n=14 980) | | | | |
| Overweight | 0.97 (0.86 to 1.08) (n=12 116) | 3.47 (3.00 to 4.01)*** (n=12 116) | 0.96 (0.79 to 1.17) (n=12 116) | 1.24 (1.10 to 1.40)*** (n=12 116) |
| Underweight | 1.22 (0.99 to 1.49) (n=11 989) | 1.79 (1.35 to 2.39)*** (n=12 116) | 1.96 (1.44 to 2.63)*** (n=12 116) | 1.28 (1.03 to 1.59)* (n=12 116) |
| Smoking | 1.10 (0.99 to 1.21) | 0.53 (0.46 to 0.61)*** | 0.47 (0.39 to 0.57)*** | 1.12 (1.01 to 1.24)* |
| Alcohol abuse | 0.38 (0.28 to 0.51)*** | 0.32 (0.21 to 0.49)*** | 0.26 (0.14 to 0.51)*** | 0.84 (0.64 to 1.10) |
| Drug use | 0.46 (0.32 to 0.69)*** | 0.50 (0.32 to 0.79)** | 0.28 (0.12 to 0.66)** | 0.71 (0.49 to 1.03) |
| Vitamin D deficiency | 0.79 (0.62 to 1.02) | 3.03 (2.28 to 3.87)*** | 2.72 (2.04 to 3.64)*** | 1.40 (1.09 to 1.78)*** |
| LARC | 0.56 (0.46 to 0.67)*** | 0.70 (0.56 to 0.89)** | 0.36 (0.24 to 0.54)*** | 0.77 (0.64 to 0.92)** |
| Emergency contraception | 0.72 (0.56 to 0.92)** | 1.86 (1.43 to 2.42)*** | 0.64 (0.41 to 1.00) | 1.19 (0.94 to 1.52) |
| TOP | 0.60 (0.40 to 0.90)* | 1.68 (1.10 to 2.56)* | 1.57 (0.94 to 2.65) | 1.14 (0.78 to 1.66) |
| Antidepressant | 0.52 (0.46 to 0.58)*** | 0.75 (0.65 to 0.86)*** | 0.51 (0.42 to 0.62)*** | 0.94 (0.84 to 1.05) |
| Antipsychotic | 0.66 (0.49 to 0.89)** | 1.48 (1.08 to 2.04)* | 0.91 (0.57 to 1.45) | 1.32 (0.99 to 1.76) |
| Valproate | 0.71 (0.41 to 1.24) | 1.29 (0.76 to 2.26) | 0.69 (0.27 to 1.75) | 1.25 (0.74 to 2.11) |
| Lithium | 0.68 (0.36 to 1.28) | 0.65 (0.35 to 1.23) | 0.69 (0.24 to 1.96) | 0.78 (0.41 to 1.45) |
| Model adjusted for IMD, SMI, valproate use and age (n=14 841) | | | | |
| Folate prescription | 0.86 (0.64 to 1.14) | 2.45 (1.84 to 3.25)*** | 1.67 (1.14 to 2.43)*** | 1.37 (1.03 to 1.82)* |
| Model adjusted for IMD, SMI and age (n=11 989) | | | | |
| Asthma | 0.53 (0.45 to 0.63)*** | 0.89 (0.73 to 1.09) | 0.75 (0.57 to 0.98)* | 1.14 (0.97 to 1.33) |
| Diabetes | 0.76 (0.50 to 1.16) | 2.26 (1.51 to 3.40)*** | 2.20 (1.34 to 3.60)** | 1.05 (0.68 to 1.62) |
| Hypertension | 0.83 (0.54 to 1.27) | 3.95 (2.67 to 5.85)*** | 1.33 (0.75 to 2.35) | 1.15 (0.74 to 1.77) |
| Epilepsy | 0.36 (0.21 to 0.62)*** | 0.63 (0.38 to 1.20) | 0.69 (0.32 to 1.52) | 1.17 (0.74 to 1.84) |
| PCOS | 0.80 (0.63 to 1.02) | 0.96 (0.69 to 1.32) | 0.97 (0.65 to 1.44) | 1.05 (0.82 to 1.34) |
| Endometriosis | 0.80 (0.53 to 1.20) | 1.46 (0.92 to 2.32) | 1.00 (0.73 to 2.32) | 1.43 (0.96 to 2.13) |
| Multimorbidity | Model adjusted for IMD and age (n=14, 980) | | | |
| Physical | 0.49 (0.34 to 0.69)*** | 1.94 (1.41 to 2.68)*** | 1.05 (0.66 to 1.68) | 0.89 (0.64 to 1.24) |
| Mental and physical | 0.98 (0.41 to 0.58)*** | 1.40 (1.17 to 1.67)*** | 0.71 (0.54 to 0.94)* | 1.19 (1.01 to 1.40)* |
| Healthcare contacts | Multivariate negative binominal regression Model adjusted for IMD, SMI and age adjusted IRR, 95% CI | | | |
| GP consultations | 0.63 (0.60 to 0.66)*** (n=14 980) | 1.26 (1.18 to 1.35)*** (n=14 980) | 0.85 (0.78 to 0.93)*** (n=14 980) | 1.09 (0.95 to 1.06) (n=14 980) |
| CMHT f2f contacts | 0.59 (0.50 to 0.68)*** (n=3399) | 0.80 (0.68 to 0.95)* (n=3399) | 0.58 (0.44 to 0.75)*** (n=3399) | 0.81 (0.70 to 0.94)** (n=3399) |
| Inpatients days | 0.88 (0.46 to 1. 67) (n=3494) | 0.75 (0.36 to 1.56) (n=3494) | 0.92 (0.33 to 2.53) (n=3494) | 1.75 (0.97 to 3.15) (n=3494) |

with white British women, but primary care remains the main provider of contraception across population groups and inequalities persist despite the source of services.[18] Finally, we also did not have cause specific mortality and further research is required to elucidate this.

**Policy implications**

Our study highlights that striking differences in multimorbidity are present in ethnic minority groups with and without mental illness at the preconception stage that may partly explain inequalities in pregnancy outcomes and mortality for black and Asian women during the perinatal period in the UK.[19] These differences suggest that there are major structural inequalities in healthcare provision in the UK. There is evidence that common mental disorders are under-recognised by primary care clinicians in women of ethnic minority groups, which has

repercussions across the health system as to how women's needs are properly assessed and prioritised by other services including maternity services. The dominance of medical model of mental illness and organisation around diagnostic criteria can also lead to inattention to people's lived experience and individual contexts, neglecting cultural differences in expressing distress and potentially leading to further retraumatisation.[20]

As part of the COVID-19 response, NHS England has issued guidance setting urgent priorities to tackle health inequalities, particularly around maternal health.[21] There is a drive to understand the needs of local populations, its health outcomes and community assets and use this understanding to plan coproduction activity to design interventions to improve equity for women and babies. There is also a push to promote personalised care and support plans as well as ensure women from ethnic minority groups are represented in peer and lay roles within local health and well-being programmes. Under the NHS Long Term Plan, maternal medicine networks will be established so that by March 2024 every woman in England with medical problems has access to specialist advice and care. In addition, maternal MH services are being developed to bring together maternity, psychology and reproductive health services for women who develop moderate–severe mental ill health from loss or trauma due to their maternity experience. These services are a substantial step forward, although there is a risk they will focus on perinatal trauma without recognition of how cumulative adversity along the life course, including racism, influences women's experiences of health services. Without embracing the wider complexity, the needs of women experiencing multiple intersecting disadvantages will continue to go unknowledge and fall between the gaps.

Our results also challenge the perspective that regular monitoring of physical health in primary care should be exclusively encouraged in people with an SMI diagnosis and indicate that there are mental and physical health needs of women in contact with MH services that are not being met by the current model of service provision. Our study emphasises the need for a more holistic approach to health promotion for women in contact with MH services, expanding the remit beyond cardiovascular disease prevention into sexual and reproductive health, including avoidance of drugs with teratogenic potential and addressing micronutrient deficiencies and substance misuse.

## Conclusions

There are striking inequalities in risk factor profile and health outcomes including mortality for women of reproductive age in contact with MH services and those of ethnic minority groups emphasising a need of culturally centred integrative models of care. More attention should be focused on identifying missed opportunities to intervene across primary and secondary MH services and closer attention should be given to how cumulative adversity along the life course, including experiences of racism, impacts on women's access to and experiences of health services.

**Acknowledgements** This work uses data provided by patients and collected by the National Health Service (NHS) as part of their care and support. We are grateful for all the support received from the National Institute for Health Research (NIHR) BRC Maudsley clinical informatics teams throughout this work.

**Contributors** RC conceived the research question and designed the study analysis plan with input from LH and SH. MP, AJ and MB performed data extraction. RC and SD analysed the data. RC wrote the first draft of the manuscript. LH, SH, MA and SD critically revised it. All authors approved the final version of the manuscript. The corresponding author attests that all listed authors meet authorship criteria and that no others meeting the criteria have been omitted. The guarantor is RC, who affirms that the manuscript is an honest, accurate, and transparent account of the study being reported, that no important aspects of the study have been omitted and that any discrepancies from the study as originally planned have been explained.

**Funding** This project has been funded by the Closing the Gap network. Closing the Gap is funded by UK Research and Innovation and their support is gratefully acknowledged (Grant reference: ES/S004459/1). The funding source had no role in project design, data analysis or writing the manuscript. This project was possible due to access to the data at the National Institute for Health and Care Research (NIHR) Maudsley Biomedical Research Centre at South London and Maudsley NHS Foundation Trust and King's College London. RC and SD receive salary support from NIHR as clinical academics (no award numbers). SH is part-funded by the NIHR Biomedical Research Centre at South London and Maudsley NHS Foundation Trust and supported by the Economic and Social Research Council (ESRC) Centre for Society and Mental Health at King's College London (ESRC Reference: ES/S012567/1). SH also receives funding from Guy's and St Thomas' Charity and Wellcome Trust (no award number). LH is an NIHR Principal Investigator and receives funding from NIHR Maudsley Biomedical Research Centre, NIHR Applied Research Collaboration South London and UKRI funded Network on Violence, Abuse and Mental Health (no award numbers).

**Disclaimer** Any views expressed here are those of the project investigators and do not necessarily represent the views of the Closing the Gap network, UKRI, NIHR, ESRC, MRC, Wellcome Trust, NHS or King's College London.

**Competing interests** All authors have completed a ICMJE disclosure form http://www.icmje.org/disclosure-of-interest/ and declare financial support from Closing the Gap Network and NIHR for the submitted work; SH receives consulting fees from the NHS Race and Health Observatory; no financial relationships with any organisations that might have an interest in the submitted work in the previous 3 years; all authors declare no other relationships or activities that could appear to have influenced the submitted work.

**Patient and public involvement** Patients and/or the public were involved in the design, or conduct, or reporting, or dissemination plans of this research. Refer to the Methods section for further details.

**Patient consent for publication** Not applicable.

**Ethics approval** Clinical Record Interactive Search (CRIS) was established in 2008 and approved by the Oxfordshire Research Ethics Committee in 2008 (reference 18/SC/0372). Approval for linkage with Lambeth DataNet was granted by Lambeth Clinical Commissioning Group and Information Governance Steering Group.

**Provenance and peer review** Not commissioned; externally peer reviewed.

**Data availability statement** Data may be obtained from a third party and are not publicly available. The anonymised dataset that underlie the results in this article is only available on request to investigators whose proposed use of data has been approved by CRIS Oversight Committee and Lambeth Clinical Commissioning Group and Information Governance Steering Group. Complete list of codes used to identify study population and outcomes can be requested from the corresponding author ( Raquel.catalao@kcl.ac.uk).

terminology, drug names and drug dosages), and is not responsible for any error and/or omissions arising from translation and adaptation or otherwise.

**Open access** This is an open access article distributed in accordance with the Creative Commons Attribution 4.0 Unported (CC BY 4.0) license, which permits others to copy, redistribute, remix, transform and build upon this work for any purpose, provided the original work is properly cited, a link to the licence is given, and indication of whether changes were made. See: https://creativecommons.org/licenses/by/4.0/.

**ORCID iDs**
Raquel Catalao http://orcid.org/0000-0003-3672-6359
Sarah Dorrington http://orcid.org/0000-0002-6462-1880
Amelia Jewell http://orcid.org/0000-0002-0887-2159
Mark Ashworth http://orcid.org/0000-0001-6514-9904
Stephani Hatch http://orcid.org/0000-0001-9103-2427
Louise Howard http://orcid.org/0000-0001-9942-744X

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
