## [Reviewer comments · BMJ Open]

ARTICLE DETAILS

TITLE (PROVISIONAL)	Ethnic inequalities in mental and physical multi-morbidity in women of reproductive age: a data linkage cohort study
AUTHORS	Catalao, Raquel; Dorrington, Sarah; Pritchard, Megan; Jewell, Amelia; Broadbent, Matthew; Ashworth, Mark; HATCH, STEPHANI; Howard, Louise

VERSION 1 – REVIEW

REVIEWER	Gomà, Marta Bruc Salut, Perinatal department
REVIEW RETURNED	31-Jan-2022

GENERAL COMMENTS	Comments to the authors I enjoyed reading the paper and I have made relatively few comments. I was very interested to see the results and the authors' discussion and interpretation of them. The large sample size and the holistic approach studying mixed variables (physical and mental health) is of great value. The present paper addresses the main objectives as stated and explains methods, tables and principal findings in detail. The conclusions are very important and actions should be taken on them once published. The wide use of abbreviations makes it a little difficult for the reader to follow the text, especially at beginning. Perhaps a list of abbreviations would help. Apart from this aspect the manuscript is interesting and easy to understand. Specifically, Page 3, line 35: Abstract: please explain the abbreviation SMI Page 4 line33: Introduction: I recommend adding "SMI" when explaining severe mental illness for easy reference later in the manuscript. The hypothesis is well written and clear. Page 5, lines 15-20: Inclusion criteria for exposed cohort: this section is a bit difficult to understand, could the authors review lines 18-20 for readability? Page 7 line 29: Results: it is not clear how the control group is selected. Could the authors add some information about the control group to diagram 1? Page 9, line 46: missing data: Around 17.4%: could you please provide a figure for the 17.4%? Page 18 lines 31-36: Policy implications Perhaps there's a need to have separate paragraphs in the policy implications: 1. 1st paragraph: From line 23 until line 31 ending with "service provision"
---

	2. 2nd paragraph: Line 32 starting with “Our study emphasizes the need for a more holistic approach” and explore this idea in greater depth. The conclusions section may benefit from the addition of stronger recommendations to policy-makers.
--	---

REVIEWER	Curry, Gwenetta The University of Edinburgh Usher Institute of Population Health Sciences and Informatics
REVIEW RETURNED	26-Mar-2022

GENERAL COMMENTS	The authors have tackled an important area of research which have been largely ignored. The results of this article will help increase awareness of the role multiple morbidities in non-pregnant women could impact future pregnancy outcomes. Black women being overdiagnosed with severe mental illness while simultaneously going undiagnosed with depression is an important point. The lack of cultural competency in mental health services can lead to patients being misdiagnosed. Previous research has highlighted bias in mental health services particularly when it comes to Black patients. The results were organized but Table 5 is a little hard to examine as there are too many variables in one table. Consider creating a separate table for the Univariate and Multivariate logistical regressions for ease of reading. I highly recommend this article for publication as its results aid in addressing inequalities in mental health services for ethnic minority women in the U.K.
---

VERSION 1 – AUTHOR RESPONSE

Reviewer: 1

Dr. Marta Gomà, Bruc Salut

Comments to the Author:

Comments to the authors

I enjoyed reading the paper and I have made relatively few comments. I was very interested to see the results and the authors’ discussion and interpretation of them. The large sample size and the holistic approach studying mixed variables (physical and mental health) is of great value.

The present paper addresses the main objectives as stated and explains methods, tables and principal findings in detail. The conclusions are very important and actions should be taken on them once published.

The wide use of abbreviations makes it a little difficult for the reader to follow the text, especially at beginning. Perhaps a list of abbreviations would help. Apart from this aspect the manuscript is interesting and easy to understand.

Specifically,

Page 3, line 35: Abstract: please explain the abbreviation SMI

We have now added severe mental illness as specification for the abbreviation in the Abstract section. We have also added an abbreviation list at the end of the manuscript in case this is thought to be required for publication (page 17).

Page 4 line33: Introduction: I recommend adding “SMI” when explaining severe mental illness for easy reference later in the manuscript. The hypothesis is well written and clear.

We have now added the abbreviation “SMI” for explaining severe mental illness in the Introduction section.

Page 5, lines 15-20: Inclusion criteria for exposed cohort: this section is a bit difficult to understand, could the authors review lines 18-20 for readability?

We have now amended inclusion criteria to read as follows: “Women aged between 15 and 40 at the start of the study window in contact with SLAM during the study window, defined as an active episode of care including a face-to-face event, were included in this study. Women were excluded from the exposed cohort if they had antenatal or postnatal Read Codes in their primary care record before window end and were registered in LDN for less than two years during in the study window. At least 1 year registration in LDN occurred after the earliest ever SLAM start date irrespective of study window.” (Methods; page 4)

Page 7 line 29: Results: it is not clear how the control group is selected. Could the authors add some information about the control group to diagram 1?

We have now edited Diagram 1 to show information about the control group. It has been uploaded as a PDF file.

Page 9, line 46: missing data: Around 17.4%: could you please provide a figure for the 17.4%?

Number of women in the control group with data missing on ethnicity (n=2,535) is now added to that paragraph (page 6).

Page 18 lines 31-36: Policy implications

Perhaps there’s a need to have separate paragraphs in the policy implications:

1. 1st paragraph: From line 23 until line 31 ending with “service provision”
2. 2nd paragraph: Line 32 starting with “Our study emphasizes the need for a more holistic approach” and explore this idea in greater depth.

The conclusions section may benefit from the addition of stronger recommendations to policy-makers.

We are grateful for the reviewer suggestion and have now considerably strengthen the policy implications section and conclusions and reformatted the paragraphs (page 15 and 16).

Reviewer: 2

Dr. Gwenetta Curry, The University of Edinburgh Usher Institute of Population Health Sciences and Informatics

Comments to the Author:

The authors have tackled an important area of research which have been largely ignored. The results of this article will help increase awareness of the role multiple morbidities in non-pregnant women

could impact future pregnancy outcomes. Black women being overdiagnosed with severe mental illness while simultaneously going undiagnosed with depression is an important point. The lack of cultural competency in mental health services can lead to patients being misdiagnosed. Previous research has highlighted bias in mental health services particularly when it comes to Black patients. The results were organized but Table 5 is a little hard to examine as there are too many variables in one table. Consider creating a separate table for the Univariate and Multivariate logistical regressions for ease of reading.

I highly recommend this article for publication as its results aid in addressing inequalities in mental health services for ethnic minority women in the U.K.

We thank reviewer 2 for their encouraging comments and we have divided Table 5 as suggested (page 12). Table 5 now only shows the multivariate logistical regressions and the univariate logistical regressions were submitted as supplementary file 2.

We hope these revisions please editor and reviewers and look forward to further feedback.

VERSION 2 – REVIEW

REVIEWER	Gomà, Marta Bruc Salut, Perinatal department
REVIEW RETURNED	07-May-2022

GENERAL COMMENTS	All the changes suggested are included. The policy implications explain in depth the need to increase awareness of the role multiple morbidities in women across the health system, and the racial implications in their attention. It is interesting to link these results with the new lines of NHS Long term plan to attend every woman in England in a holistic approach when explaining "maternal mental health services are being developed to bring together maternity, psychology and reproductive health services for women who develop moderate–severe mental ill health from loss or trauma due to their maternity experience" The authors have addressed a very important area of research widely ignored. I highly recommend this article for publication as its results address the important area of inequalities in mental health services for ethnic minority women in the U.K and their implications.
--